# Exploring the Impact of Workplace Violence in Urban Emergency Departments: A Qualitative Study [note 1]

**DOI:** 10.3390/healthcare13060679

**Published:** 2025-03-20

**Authors:** Brendan Lyver, Brendan Singh, Nathan Balzer, Manu Agnihotri, Jennifer Hulme, Kathryn Chan, Rickinder Sethi, Charlene Reynolds, Jennifer Haines, Robert Whiteside, Marc Toppings, Christian Schulz-Quach

**Affiliations:** 1University Health Network, Toronto, ON M5G 1X6, Canada; brendan.lyver@uhn.ca (B.L.); brendan.singh@uhn.ca (B.S.); manu.agnihotri@uhn.ca (M.A.); rickinder.sethi@uhn.ca (R.S.);; 2Department of Family Medicine, McMaster University, Hamilton, ON L8S 4L8, Canada; 3Department of Family and Community Medicine, University of Toronto, Toronto, ON M5S 1A1, Canada; 4Temerty Faculty of Medicine, University of Toronto, Toronto, ON M5S 3K3, Canada; 5Institute of Health Policy, Management, and Evaluation, University of Toronto, Toronto, ON M5T 3M6, Canada

**Keywords:** workplace violence in health care, emergency medicine, pandemic recovery, qualitative research

## Abstract

**Background:** Workplace violence (WPV) in healthcare, particularly in emergency departments (EDs), is a growing and complex issue that significantly impacts healthcare providers (HCPs). Following the COVID-19 pandemic, the rates of WPV have increased globally, contributing to increased burnout, reduced morale, and heightened challenges in staff retention within EDs. **Objective:** This study aimed to explore HCPs’ perspectives on WPV in EDs. The insights gathered were intended to inform targeted interventions in a quality improvement initiative focused on addressing WPV in our healthcare institutions. **Methods:** A qualitative study involving semi-structured interviews was conducted with 52 HCPs across two urban EDs at a multi-site academic health center. Participants from various roles and shifts shared experiences related to safety, leadership, training, and security. Interviews were recorded, transcribed, anonymized, and thematically analyzed to identify key themes surrounding WPV in EDs. **Results:** The thematic analysis identified four main themes, including (1) Violence is Part of the Job, (2) Leadership Dynamics regarding WPV, (3) Disconnect Among ED Staff During WPV Response, and (4) Enhancing Systems and Culture for Effective WPV Management. These themes provide a comprehensive view of HCPs’ experiences and offer actionable recommendations for healthcare organizations seeking to address WPV. **Conclusions:** The study provides valuable qualitative insights into WPV in EDs, emphasizing the importance of addressing cultural, structural, and procedural gaps. These findings can guide the development of more supportive and effective strategies to create a safer environment for ED staff. Further rounds of interviews are planned post-intervention to assess changes in HCPs’ perceptions and experiences.

## 1. Introduction

### 1.1. Problem Formulation

Workplace violence (WPV) in healthcare is a growing challenge that poses significant risks to patient care and healthcare provider (HCP) well-being and burnout [1,2]. Since the COVID-19 pandemic, healthcare institutions globally have experienced an increase in WPV, particularly in emergency departments (EDs) [3,4]. WPV has multiple definitions [5]; however, for the purpose of this study, we defined WPV as any act of physical, verbal, psychological, and/or sexual violence committed by patients or visitors against HCPs [1,6]. Additionally, the term HCPs in this study encompasses all ED staff interviewed, including physicians, nurses, allied health professionals, ward clerks, housekeepers, and security personnel, all of whom contribute to patient care and safety in the ED.

WPV in healthcare is a complex and multifaceted issue influenced by numerous risk factors, which can be categorized as clinical, environmental, organizational, societal, and economic [7]. Clinical risk factors are associated with the patient population, such as agitation in dementia patients or delirium in post-anesthesia patients [1,7]. Environmental risk factors include inadequate escape options, blind spots, and unsecured furniture [1,7]. Organizational risk factors encompass a culture of underreporting, lack of leadership regarding WPV prevention, and lack of education regarding WPV prevention [1,7]. Examples of economical risk factors include lack of funding towards WPV prevention and inadequate staffing to environmental risk ratios [1,7]. Societal risk factors involve global events or conflicts, under-resourced communities, and societal perspectives of vulnerable populations [1,7]. Societal risk factors further amplify the complexity of WPV, as they extend beyond the control of healthcare institutions. These include the broader impact of global events and conflicts, as well as systemic issues such as insufficient community resources, which often result in EDs becoming the point of last resort for vulnerable populations [1,7].

WPV has far-reaching consequences, affecting not only HCPs but also patients’ (chosen) family members, visitors, and healthcare institutions [1]. For HCPs, WPV can result in physical harm, psychological distress, increased burnout, and declining mental well-being [8]. These challenges impair job performance and reduce the quality of patient care, ultimately placing patients at risk [9,10]. Additionally, exposure to WPV may influence how HCPs respond to agitated patients, potentially escalating conflicts rather than de-escalating them, which can further perpetuate violence and negatively impact all involved, including the healthcare institution [11]. The rising prevalence of WPV is linked to decreased job satisfaction and increased turnover rates, particularly in high-stress environments like the emergency department, further straining healthcare organizations [8,10].

Addressing WPV effectively requires a systems-level approach that considers its multifaceted nature and is guided by a data-driven strategy [1]. Quantitative organizational data may serve as the foundation for these efforts, offering essential insights into the prevalence, frequency, and patterns of WPV incidents and informing areas that require targeted intervention [12]. However, these data often reveal only broad trends, lacking the nuance needed to tackle the complex and multifaceted nature of WPV effectively [1]. To bridge this gap, a bottom-up approach incorporating qualitative studies is essential, as qualitative data provide a deeper understanding of the lived experiences and challenges faced by HCPs, insights that quantitative methods alone cannot capture [1,12]. Prior qualitative studies on WPV in healthcare and ED settings have deepened our understanding of WPV from the perspectives of HCPs, identifying key contributing factors and unmet needs [13,14,15]. HCPs have reported that WPV is often linked to encounters with patients experiencing mental health conditions or substance use disorders, as well as the negative public perception of nurses [14,15]. These studies also highlight the need for organizational initiatives, such as stronger policies, appropriate staffing levels, and comprehensive education programs to address WPV effectively [13,14,15]. Additionally, HCPs have emphasized the importance of enhanced security measures, including personal panic alarms, improved surveillance systems, and better documentation of violent incidents [13,15]. A strength of some studies is their use of multiple locations and diverse HCP roles to enhance the applicability of findings, which we replicated in our study [13,15]. However, generalizability remains a limitation, as findings may be specific to an institution. Nevertheless, recurring themes across studies, including the role of mental health and substance use in WPV, the impact of inappropriate staffing levels, and the need for improved security measures, have strengthened the broader understanding of WPV and the generalizability of past studies’ findings [13,14,15]. Concurrently, institutionally unique factors, such as differences in hospital policies, variations in security protocols, and the contrast between private and public healthcare systems, where financial costs may intensify WPV, can lead to new insights and contribute to a more complex understanding of WPV, underscoring the need for tailored interventions [13,14,15]. To our knowledge, this is the first published qualitative study on WPV in Canadian EDs, providing an opportunity to reinforce existing findings while contributing new context-specific perspectives.

Additionally, engaging healthcare providers in such studies is crucial, as their frontline perspectives can uncover underlying causes of WPV and inform mitigation strategies [13]. Furthermore, longitudinal qualitative data collection enables the identification of changes over time, validating and enriching trends observed, ensuring interventions remain impactful and adaptive [1]. While this manuscript serves solely as a baseline for understanding WPV in the healthcare setting, it is part of a larger, ongoing longitudinal quality improvement (QI) initiative. Future publications will explore the longitudinal aspects and identify changes over time to validate and enhance observed trends.

### 1.2. Purpose

This study aimed to explore ED HCPs’ lived experiences with WPV and deepen our understanding of WPV within our healthcare institution. Specifically, our research questions ask (1) How do emergency department healthcare providers experience and perceive WPV? (2) What changes do they believe are necessary to address WPV effectively? We examined HCPs’ sense of support and safety, perceptions of security, and personal expectations for necessary changes to address WPV. Employing a bottom-up qualitative approach, we generated institution-specific insights that will guide the development of actionable interventions to enhance both the workplace environment and ED staff well-being.

## 2. Materials and Methods

### 2.1. Qualitative Approach and Research Paradigm

This study was conducted as part of a larger QI initiative addressing WPV at the University Health Network (UHN), a multi-site academic health science center in Toronto, ON, Canada [1]. As one of 12 subprojects, we employed a phenomenological approach to explore HCPs’ lived experiences of WPV in EDs. Phenomenology was selected for its ability to capture the essence of shared experiences, focusing on how HCPs describe and process WPV. This approach was chosen over interpretative phenomenological analysis, because our goal was to identify common experiential themes to inform our QI initiative rather than conduct a deep psychological analysis of individual meaning-making, which is more suited to interpretative phenomenological analysis. This study employed a qualitative research design, guided by the Standards for Reporting Qualitative Research (SRQR) guidelines [16], to explore experiences and perceptions of WPV among HCPs in two urban EDs of a multi-site academic health science center.

### 2.2. Researcher Characteristics and Reflexivity

We conducted the study as an interdisciplinary team comprised of two staff psychiatrists, a Security Director, a Chief Legal Officer, a Project Manager, a Research Analyst, and a Research Assistant. The Research Analyst led the study and conducted all qualitative interviews, as a non-clinician with no prior experience in the EDs or any preexisting relationships with their staff, ensuring a neutral position during data collection. The Research Analyst was a master’s student in the Psychology and Neuroscience of Mental Health program with one year of experience in QI projects in healthcare and a foundational understanding of WPV in healthcare. Thematic analysis was conducted collaboratively by the Research Analyst and a Research Assistant. The Research Assistant was an undergraduate student in Psychology with no prior experience in clinical settings, QI projects, or the ED environment but possessed a fundamental knowledge of WPV in healthcare. This combination of foundational knowledge and novice perspectives supported a balanced and thorough analysis, minimizing potential biases and enhancing the study’s rigor.

Additional team members contributed to the iterative development of the semi-structured interview guide and offered critical feedback during the thematic analysis process, enhancing the depth and validity of the findings. This included two staff psychiatrists with clinical experience in the ED and Psychiatric Emergency Service Unit, one of whom also served as the Medical Director of WPV prevention. Together, they contributed clinical insight to the study’s development and provided real-world experience to validate the emerging themes. Additionally, the Security Director and Security Project Manager were included, and while they lacked clinical experience and direct work in the ED, their expertise in security and WPV contributed to the development of the interview guide and ensured that the questions targeted areas of interest. Lastly, the organization’s Chief Legal Officer contributed to the development of the study; the Chief Legal Officer also had no clinical experience or direct experience in the ED but provided organizational oversight at an executive level.

### 2.3. Context

We conducted this study at UHN’s two urban EDs, which collectively served 158,000 patients in 2022. Following the COVID-19 pandemic, these EDs experienced a significant increase in WPV incidents, and the reported rates rose from 0.43 to 1.15 incidents per 1000 ED visits (*p* < 0.0001); this reflected global trends [1].

In response to this alarming rise, the UHN Department of Security Operations formed a QI team in November 2022 to address WPV in healthcare settings. One of the team’s initial initiatives was to collect data directly from frontline ED staff to guide the development of a comprehensive QI strategy and ensure staff felt heard and supported in addressing and managing WPV.

### 2.4. Sampling Strategy

This study used a convenience sampling approach to accommodate the demanding nature of HCPs’ work in the ED. We conducted interviews on-site during staff shifts to maximize participation while minimizing disruption to personal time. However, this strategy introduced the limitation of a reliance on staff availability during shifts, which may have influenced both the representativeness of the sample and the completeness of the data.

We included participants with a minimum of three months of ED experience who were willing to share their WPV experiences. We excluded HCPs on leave, those with less than three months of ED experience, and agency staff.

We continued sampling until reaching data saturation, which the researcher determined when no new relevant information or themes emerged from the interviews. Throughout the process, the researcher evaluated whether additional data would contribute unique insights to the study’s objectives. Once further interviews no longer added new perspectives on WPV, we deemed additional sampling unnecessary.

### 2.5. Ethical Issues Pertaining to Human Subjects

The study received formal exemption from the Research Ethics Board and approval from the organization’s Quality Improvement Review Board (QI ID: 22-0499). We provided all participants with a detailed consent form outlining the QI project purpose, objectives, procedures, potential risks, and their right to withdraw at any time without consequence (Appendix A). To protect participants’ confidentiality, we stored all data on a secure server only accessible to the QI team.

### 2.6. Data Collection Methods

Participants were initially informed about the study during daily huddles held in the EDs, and those interested were invited to participate in on-site, in-person interviews during their shifts. All interviews were conducted in a private interview room within the ED to maintain confidentiality. To accommodate ED staff, virtual interviews were also offered via Microsoft Teams. Interviews lasted between 30 min and 1.5 h, depending on participant availability and the depth of the discussion.

### 2.7. Data Collection Instruments and Technologies

A semi-structured interview guide was collaboratively developed with the QI team, organizational WPV experts, and ED leadership, undergoing iterative refinement to ensure its relevance and applicability to WPV in healthcare (Appendix A). Vignettes from *Code White: Sounding the Alarm on Violence Against Healthcare Workers*, a book on the WPV epidemic in Canada [7], were incorporated to prompt realistic discussions, with all receiving prior approval. We included specific questions into the interviews to collect quantitative data on HCPs’ perspectives on intervention effectiveness and inform our QI initiative. The team selected the interventions through discussions, as well as suggestions from *Code White: Sounding the Alarm on Violence Against Healthcare Workers,* to ensure their relevance to WPV prevention efforts [7]. The interviews were recorded using Zoom or Microsoft Teams, depending on the format. A Shure SM57 microphone and Steinberg UR12 interface were used to record in-person sessions.

### 2.8. Units of Study

The interviews were conducted in December 2022 with participants (n = 52) representing HCPs from two urban EDs, spanning a broad range of professions (n = 9) and capturing the diverse experiences of patient-facing staff in the ED (Table 1). To ensure diverse perspectives, both day shift (n = 37) and night shift (n = 15) staff were included, as well as a mix of women (n = 39) and men (n = 13).

### 2.9. Data Processing

An auto-transcription software was used to transcribe the recordings, and all transcripts were anonymized and reviewed for accuracy. MAXQDA 24, Version 24.1.0 qualitative analysis software was used for data organization and management. Due to technical issues, including recording and downloading errors, data from six interviews were lost, resulting in a final dataset of 46 interviews for analysis.

### 2.10. Data Analysis

We conducted a thematic analysis following the framework outlined by Kiger and Varpio and used inductive coding [17]. We chose inductive coding for its ability to generate themes directly from the data, ensuring that the findings are grounded in HCPs’ experiences with WPV in EDs rather than being influenced by predefined assumptions, as in deductive coding approaches [17]. This data-driven approach allows for a structured yet adaptable method to capture the complexity of WPV in healthcare settings. Kiger and Varpio’s approach to thematic analysis includes the following steps: (1) Familiarizing Yourself with the Data, (2) Generating Initial Codes, (3) Searching for Themes, (4) Reviewing Themes, (5) Defining and Naming Themes, and (6) Producing the Report [17]. This approach is closely aligned with the coding reliability thematic analysis described by Braun and Clarke [18]. Aspects of this approach utilized include interrater reliability to ensure consensus among multiple coders and enhance rigor, the development of themes that offer overviews of responses, and the identification of supporting evidence for these themes [18].

The Research Analyst and Research Assistant collaborated on the analysis, with the latter offering an external perspective to enhance the trustworthiness of the theme development and mitigate potential bias arising from prior familiarity with the data. Each researcher independently conducted steps 1 through 4; after which, they defined and named the themes through iterative discussion. Differences in interpretation were explored and discussed until consensus was reached, ensuring consistency and enhancing the analytical rigor.

In addition to conducting a thematic analysis, we extracted quantitative data from the interviews. Structured yes-or-no questions were incorporated into the interviews to gather direct feedback on specific interventions of interest for QI. This provided a targeted assessment of HCPs’ perspectives on intervention effectiveness. To further support our thematic analysis findings, we examined the frequency of the themes and subthemes across the interviews. The quantitative data extracted from this study will serve to complement the findings of the thematic analysis but will not undergo further in-depth analysis.

### 2.11. Techniques to Enhance Trustworthiness

To enhance the trustworthiness and credibility of the thematic analysis, we conducted interrater reliability to ensure consistency in coding and theme development. Two researchers independently analyzed the data and compared their findings to ensure that the themes identified were both reliable and accurately represented the participants’ experiences.

Additionally, we reviewed and discussed all emerging themes with two ED HCP members, two psychiatrists, and a director from the Center for Mental Health. This process of member checking and peer debriefing helped to validate the findings and ensured that the interpretations aligned with the lived experiences of the participants and the expertise of those working in the field.

## 3. Results

### 3.1. Synthesis and Interpretation

Following a thematic analysis and member checking, we identified four main themes, including (1) Violence is Part of the Job, (2) Perceived Leadership Dynamics regarding WPV, (3) Disconnect Among ED Staff During WPV Response, and (4) Enhancing Systems and Culture for Effective WPV Management. Table 2 summarized these themes, accompanied by quotes that substantiated the findings from the thematic analysis.

### 3.2. Violence Is Part of the Job

The first main theme explored the perspectives of ED HCPs on WPV. ED HCPs reported widespread desensitization to the WPV they experience in their daily work. Although personal safety remains a concern, HCPs often prioritize patient care over their own well-being. WPV is described as an unavoidable aspect of their profession. This mindset significantly influenced their sense of well-being, both within and beyond the workplace. We identified four subthemes: (1.1) Desensitization to WPV in the ED, (1.2) A Prevailing Sense of Hopelessness, (1.3) Identity-targeted WPV, and (1.4) Moral Injury and Moral Distress.

Desensitization to WPV: The first subtheme emphasized that high WPV occurrence rates among ED HCPs led to the acceptance that WPV is an inevitable aspect of their job. This normalization is reinforced by the regularity of WPV incidents, including Code White events that often go unreported or are not formally called this. Additionally, the absence of follow-up or debriefing after such events contribute to a culture where WPV is normalized rather than addressed as a serious concern requiring organizational attention. Verbal WPV, in particular, was so frequent that it was often described as unavoidable and part of the job, further desensitizing ED HCPs to the abuse that they routinely encountered while in the workplace.

Prevailing sense of hopelessness: ED HCPs expressed skepticism about the effectiveness of potential WPV interventions. They described the ED as a fast-paced, high-pressure environment where high patient volumes and complex, unpredictable patient populations created seemingly insurmountable barriers to addressing WPV. Staff highlighted particular challenges in managing patients with substance use or mental health conditions. They anticipated that interventions will prove ineffective due to patients’ lack of responsiveness or coherence. Consequently, they viewed measures like awareness posters and educational initiatives as unlikely to create meaningful change. The perceived absence of robust organizational initiatives or communication about WPV reinforced these feelings of hopelessness.

Identity-Targeted WPV: ED HCPs experienced WPV, particularly verbal abuse, that often targeted aspects of their identity. This included acts of misogyny, homophobia, and racism. HCPs who identified as men noted that patients and visitors frequently assumed they were doctors and treated them with greater respect, whereas HCPs who identified as women were often presumed to be nurses and faced poorer treatment as a result. The impact of these identity-based attacks varied among HCPs, as some described them as deeply distressing and traumatic, while others, though frustrated, felt desensitized due to the frequency of such incidents.

Moral injury and moral distress: WPV in EDs contributed to job-related burnout, diminished mental health and well-being, and reduced self-efficacy among HCPs, and this highlighted the emotional and physical toll of these events. Acceptance culture within the ED often discouraged open discussions about WPV. This left HCPs to process traumatic WPV events on their own time with their own resources. Furthermore, ED HCPs prioritized patient care over their own safety and well-being due to feeling it was their duty and an expectation of the job. This created a challenging dynamic where providing high-quality care became increasingly difficult while managing the threat of WPV.

### 3.3. Leadership Dynamics Regarding WPV

ED HCPs expressed mixed perceptions of leadership support in addressing WPV. Although many staff members reported feeling adequately supported by ED management, there was a sense of being overlooked by the broader organization. This perceived lack of initiative from the organization contributed to feelings of frustration and neglect among staff. The subthemes include (2.1) Support from ED Management and (2.2) Perceived lack of Organizational Initiative with regards to WPV.

Support from ED Management: ED HCPs spoke highly of ED management due to their ongoing support regarding WPV. Staff noted that management ensured they are accessible and approachable, offering support to staff in addressing challenges related to WPV. Additionally, ED managers were recognized for their responsiveness to staff concerns and their visible presence following severe WPV incidents, which provided reassurance and reinforced a sense of support during critical times.

Perceived Lack of Organizational Initiative: In contrast, ED HCPs expressed frustration with the perceived lack of initiative from the organization in addressing WPV. Many reported that hospital management failed to actively support staff, citing a lack of acknowledgment of WPV incidents and the physical absence of leadership following severe events. Staff perceived that submitted incident reports disappeared into a void, and they felt that there was a lack of follow-up from leadership, leading to staff feeling dismissed. Perceived limited communication on WPV prevention and inadequate investment in intervention and training programs further reinforced these concerns. The discontinuation of in-person training after the beginning of the COVID-19 pandemic, combined with the evaluation that onboarding training was minimal left staff with a feeling of being forgotten and unsupported. Additionally, staff felt that a lack of resources, such as panic buttons and metal detectors, further demonstrated the organization’s lack of initiative.

### 3.4. Perceived Disconnect Among ED Staff During WPV Response

HCPs expressed a strong sense of camaraderie with their colleagues, emphasizing team unity and the absence of an occupational hierarchy. While HCPs recognized security as an essential part of the medical team, security personnel felt disconnected from the broader healthcare team, a sentiment echoed by ED staff. Additionally, a lack of awareness of other professions’ roles and limitations led to confusion during WPV and Code White incidents. The subthemes that emerged were (3.1) Interactions between members of the healthcare team in the ED and (3.2) Role Confusion during WPV Incidents.

Interactions between members of the healthcare team in the ED: HCPs reported strong teamwork as a key strength, with no clear positional hierarchy, especially during WPV incidents. They attributed higher WPV rates among nurses to more frequent patient interactions rather than occupational hierarchy. Staff linked the high prevalence of WPV against nurses to time spent with patients, as well as that patients may be less likely to behave violently toward physicians, nurse practitioners, and physician assistants because they rely on them for specific needs.

Role Confusion during WPV Incidents: HCPs and security guards maintained critical relationships for effective de-escalation. HCPs acknowledged security’s importance in ensuring safety, and they noted that security guards actively patrolled the ED, maintained a calm demeanor, used effective de-escalation methods, and assisted in implementing protective measures to ensure patient safety during care provision. Additionally, ED HCPs highlighted that the presence of uniformed team members can positively influence patient behaviour. While HCPs valued security’s role in de-escalation, describing guards as integral members of the medical team and contributing to their sense of safety (Table 3), security guards often felt like outsiders, disconnected from the healthcare team and underappreciated.

Despite the sense of camaraderie among HCPs, there was confusion regarding the role delineation during a Code White or WPV responses. Miscommunication led to uncertainty about each person’s responsibilities and areas of control, particularly between HCPs and security guards. This confusion was heightened by unclear role limitations between interdisciplinary team members and discord regarding the most appropriate de-escalation techniques.

### 3.5. Enhancing Systems and Culture for Effective WPV Management

The final theme focused on areas for improving systems and culture to enhance the prevention of WPV in UHN EDs. HCPs reported mixed feelings about their safety and highlighted gaps in the current security measures, de-escalation knowledge, and organizational support. The subthemes include (4.1) Current Measures and Feelings of Safety, (4.2) Areas for Improvement and Prospective Measures, (4.3) Need for Education, (4.4) Streamlined Code White Process, and (4.5) Environmental Concerns.

Current Measures and Feelings of Safety: While current security measures like guards’ presence and behavioural safety assessments (BSAs) are considered effective, HCPs reported gaps in de-escalation knowledge and response strategies. A stronger organizational culture and improved management support were seen as critical for enhancing staff safety and confidence in handling WPV.

Areas for Improvement and Prospective Measures: ED HCPs identified several key areas for improvement in WPV management. Staff expressed feeling severely understaffed, which hampered their ability to ensure both patient and personal safety. Additionally, there was a strong call for additional security guards to support the ED team. The proposed interventions, such as body-worn cameras, improved BSAs with de-escalation strategies tailored to the patient, and personal panic buttons, were viewed as critical to improving staff safety, increasing confidence in WPV management and fostering a greater sense of organizational support. Furthermore, ED HCPs called for policy changes regarding WPV and Code White responses, requesting clear protocols and follow-up after incidents. The professional standards and training of security staff also need to be improved. Proactive strategies, such as building rapport with patients, checking in with staff, and maintaining a visible and accessible presence to ED HCPs, were viewed as essential for improving the overall WPV management approach.

Need for Education: A lack of ongoing WPV prevention training, particularly after onboarding, was a prevalent concern among staff. HCPs emphasized the need for comprehensive, in-person de-escalation training to bridge the knowledge gap between senior and newer staff. Additionally, HCPs reported the need for training to address the stigmatization of patients with mental health or substance use issues. Furthermore, it was emphasized that ongoing refreshers would be necessary to ensure the effectiveness of training.

Streamlined Code White Process: ED HCPs highlighted the contrast between Code White and Code Blue responses, noting that Code Blues (i.e., medical emergency response) followed a more streamlined and systematic process. In contrast, Code White responses were perceived as ambiguous and, at times, chaotic. Staff identified key differences that could improve Code White responses, including a more structured response, clearly defined roles and steps, debriefs following incidents, and case reviews with organizational feedback on how to improve or prevent future occurrences. ED HCPs believed these changes would enhance the de-escalation efficacy and reduce confusion regarding roles during Code White incidents.

Environmental Concerns: ED HCPs expressed concerns about the physical environment, acknowledging that, while these issues are difficult to change, they would enhance their feelings of safety. Key concerns include panic buttons that are either not accessible due to location or not always functional and the physical layout of rooms and floors that creates blind spots and limits exits. Staff noted that beds can obstruct access to panic buttons or exits and doors near the Rapid Assessment Centre are not locked 24/7, allowing easy access for unwanted visitors. Additionally, staff identified weapons of opportunity, such as IV poles, scalpels, and needles, which are sometimes left out in the open.

### 3.6. Quantitative Findings

Several questions were posed to gather feedback on potential WPV prevention interventions, with the goal of evaluating their level of support from ED staff. The results offered valuable insights into staff perspectives on effective WPV prevention interventions (Table 4).

Strongly supported interventions included personal panic alarms for ED staff (n = 43), flagging patients with behavioural issues (n = 42), Code White simulations (n = 41), and routine check-ins with staff (n = 40). Personal panic alarms were particularly favored, as they provided HCPs with the ability to call for help regardless of where they were, whereas stationary panic buttons on walls were not always in reach. Flagging patients with behavioural issues was seen as a valuable tool for increasing HCPs’ awareness, enabling proactive measures to ensure safety and manage interactions effectively. Code White simulations were strongly supported, as staff described Code Whites as chaotic and hoped for more structure. Lastly, HCPs appreciated the interviews as an opportunity to be heard by the organization and expressed a desire for routine check-ins and communication with leadership to ensure their voices were acknowledged.

Conversely, interventions such as wearable devices for security guards (e.g., body cameras) (n = 19), environmental indicators for harm reduction (n = 27), and an additional security guard in the ED (n = 29) received less support. Concerns regarding body cameras were primarily related to issues of patient confidentiality, with staff expressing reservations about their potential misuse, particularly regarding the selective use of cameras in situations that may portray security personnel in a favorable light. Similarly, environmental indicators were viewed with skepticism, as staff believed that agitated patients would likely disregard these signals and that such indicators alone would be insufficient to effectively mitigate violence. Lastly, the lower support for an additional security guard was attributed to participants’ belief that more than one additional guard was necessary.

## 4. Discussion

### 4.1. Integration with Prior Work, Implications, Transferability, and Contributions to the Field

This study enhanced our understanding of WPV in EDs by exploring the perspectives of ED HCPs. Our analysis revealed a culture where WPV is normalized, provided insights into the dynamics between ED HCPs and leadership, and highlighted the interactions between roles in the ED. Moreover, it identified actionable changes necessary to address WPV. These qualitative findings complement quantitative data by addressing gaps and offering a foundation for designing a comprehensive, systems-based strategy to mitigate WPV.

Addressing WPV in healthcare requires an urgent and complex approach, considering numerous factors at the systems level [1]. Effective WPV management necessitates a comprehensive strategy [1,19,20]. Designing such an approach requires defining the problem and finding the data [1]. While top-down methods using organizational data offer valuable insights, they are often limited by the underreporting of WPV by staff, which can skew metrics and misrepresent the true extent of the issue [12,21,22]. This underscores the necessity of qualitative approaches like this study to build a more accurate and actionable understanding of WPV.

Building on these insights, the remainder of this discussion investigates the key findings of this study, examining their implications in the context of the existing literature. By situating these results within broader research, we aim to highlight how the perspectives of ED HCPs deepen our understanding of WPV and inform actionable strategies for addressing this pervasive issue.

This study identified the normalization of WPV among ED HCPs, a trend that has been consistently observed in the existing literature [14,15]. Contributing factors to WPV identified in this study and previous research include ineffective organizational management, increased workload pressures, and high patient volumes [14,15]. The alignment of these findings with prior research across diverse healthcare settings suggests that the normalization of WPV may be a widespread issue and increases the generalizability of this theme.

This study also explored the resulting moral distress and moral injury experienced by ED HCPs, driven by the perceived ethical and professional duty to prioritize patient care, even at the expense of their own safety and well-being [23]. An example of moral injury is evident in participants’ responses when asked whether they would press charges against a patient who had physically assaulted them. While many acknowledged that such actions constituted a crime, they expressed hesitation, feeling that pursuing legal action conflicted with their responsibilities as caregivers or that their organization would disapprove, as HCPs should prioritize the patient’s needs over their own well-being. The resulting moral injury contributed to burnout, post-traumatic stress disorder, and other trauma-related conditions, ultimately compromising both the mental health of HCPs and the quality of care they provided [24].

Similar to our findings, the normalization of WPV has been widely documented in the literature, with studies reporting its role in aggravating moral injury, negatively impacting care delivery, job satisfaction, and increased HCP turnover rates [15,25,26]. These findings highlight the urgent need for comprehensive interventions to mitigate WPV and challenge the pervasive belief that violence is an inevitable part of the job.

Effective organizational support and communication are crucial in mitigating the impact of WPV; however, this study found that ED HCPs felt the organization did not prioritize addressing WPV in the ED, as participants had limited awareness of initiatives being implemented within the organization. Past studies have highlighted that limited perceived organizational support contributed to HCP burnout [27,28]. Additionally, research indicates that perceived organizational support is positively correlated with tenure, suggesting that high turnover rates in EDs may reinforce the perception that the organization does not prioritize staff safety [29]. Despite these concerns, ED HCPs often viewed ED leadership as supportive due to their visible presence and actions within the department. However, they were less aware of the organization’s broader influence, as shown when the addition of a security guard was attributed to ED leadership rather than the organization. This reflects a fundamental attribution error [30], where individuals overemphasize internal factors, such as leadership actions, while underestimating external organizational roles. To increase ED HCPs’ awareness of organizational support, organizations must implement a clear and cohesive communication strategy [1]. Transparent communication ensures staff recognize broader organizational efforts to address their concerns, fostering cohesion, enhancing buy-in, and strengthening trust [1,31,32]. Additionally, leadership can adopt ED leadership practices, such as regular physical presence and open communication channels, to reinforce support and responsiveness to WPV concerns.

While the previous literature has identified that hierarchical dynamics in EDs can contribute to certain roles, such as nurses, experiencing more WPV than others [7], respondents felt this was not the case in UHN EDs. Instead, ED HCPs reported a strong sense of mutual respect and camaraderie across professional roles. However, staff acknowledged that societal views of gender might contribute to the higher prevalence of WPV among nurses, particularly women. Male nurses noted feeling less likely to be targeted, as patients and visitors often assumed they were doctors, reflecting societal norms and gendered assumptions [33].

Additionally, ED HCPs with diverse intersectional identities reported experiencing verbal WPV directed at aspects of their identity, highlighting the influence of unconscious biases and intersectionality in WPV [33,34,35]. This is an example of a societal contributing factor to WPV that is difficult to address, as it is outside the scope of a healthcare organization [7]. Nevertheless, organizations can support ED HCPs by offering resources such as counseling, leave of absence, and structured debriefs, alongside regular check-ins, to provide ED HCPs with support [36,37,38]. As well as a multipronged approach to ameliorate WPV through WPV prevention training, environmental factors and organizational strategies such as policy changes can create a safer workplace for ED HCPs [1,19,37].

The presence of security in EDs is widely recognized as a crucial factor in enhancing workplace safety and preventing WPV [13,39], a finding reinforced by the themes identified in this study. However, our analysis uncovered a notable disconnect between HCPs and security personnel, an issue that remains underexplored in the existing literature. While HCPs value the sense of safety provided by the guards’ presence and de-escalation skills, security staff expressed feeling excluded from the medical team. Both groups identified a lack of role clarity during WPV events as a significant contributor to this disconnect. Factors such as siloed work culture and mistrust between colleagues may worsen this dynamic [40]. Additionally, high turnover rates in healthcare and ineffective team-building practices further hinder collaboration and reinforce feelings of isolation [41]. Respondents highlighted the urgent need for role clarification as part of a broader WPV prevention education program and a coordinated Code White response structure. The existing literature points to a lack of practical, coordinated WPV response guidelines in healthcare systems, often due to the ambiguous nature of WPV events, which makes it challenging to implement consistent behavioural techniques across diverse scenarios [13].

In relation to effective organizational and security measures to address WPV, another thematic analysis study highlighted similar effective security measures to our analyses. Key themes related to security interventions found that, while alarm systems increased a personal sense of safety, their placement and inaccessibility were noted as limiting factors to their use [13]. Additionally, behavioural reporting systems were considered effective measures to prepare HCPs before treating patients but presented limited descriptions of triggers and effective de-escalation techniques, both of which were barriers similarly found in our analyses [13]. The reported WPV response structure often took the form of a designated responder or de-escalation being conducted in pairs; however, evidence-based guideline procedures for response efforts were not available or not regularly adhered to [13].

The prior literature has additionally highlighted staff presence and security patrols as a crucial resource for de-escalation, in turn increasing personal feelings of safety among HCPs [42,43]. Our analyses found that staff felt most vulnerable when shifts were understaffed or stretched thin when covering breaks or responding to other codes. Additionally, the ambiguity of WPV events results in senior staff generally taking leading roles in de-escalation due to past experience [44]. While experienced staff assume governing roles in de-escalation and use acquired techniques to aid de-escalation, there exists an increased need for generalized de-escalation skills and training in order to prepare junior and less experienced staff for potential events [44,45]. The literature has found that junior nurses report not being adequately prepared for WPV incidences, and improved evidence-based de-escalation training is needed, specifically at the onboarding and post-secondary level [46].

Peer support workers are emerging as a valuable resource in reducing WPV in EDs and have been working in the UHN EDs since 2020 in a hospital–community partnership with The Neighbourhood Group. These individuals, who have lived experience with homelessness, substance use, or mental health challenges, provide specialized care and support for patients facing similar issues. By building trust and rapport with high-risk patients, peer support workers can help de-escalate potentially volatile situations before they escalate into violence. Their ability to relate to and communicate effectively with marginalized populations helps bridge gaps in the understanding between patients and clinical staff, potentially reducing frustration and aggression. While we did not address the role of peer support workers in this study, early findings presented at the Canadian Association of Emergency Physicians (CAEP) 2024 Conference suggested that peer support workers have a positive impact on patient care and staff experiences in the ED. The study found that peer support workers were able to de-escalate more than half (60.6%) of escalating situations with patients. Additionally, in over one-third (38.9%) of all patient interactions, peer support workers were able to share new information about the patient with the healthcare team, which could be crucial for providing appropriate care and preventing misunderstandings. Furthermore, the qualitative data from the study revealed that peer support workers contributed to improvements in care transitions, particularly during the discharge period, and created space for a trauma-informed approach to care delivery. Their ability to establish empathy and build trust between patients and the broader healthcare team was emphasized as a key strength of their role. Further studying the role of peer support workers in addressing WPV as part of multidisciplinary teams is an important priority for future research.

A more challenging factor to address is the environmental layout of the ED, which poses unique obstacles in mitigating WPV. Structural changes to existing facilities are often difficult to execute and costly, limiting the potential for improvements. The literature and our analysis highlighted the importance of having clear sightlines, designated secure spaces, and accessible exits [7,47]. While structural changes to existing EDs may not be feasible, integrating feedback from HCPs during the design phase of new EDs can significantly improve safety protocols and contribute to a safer work environment for HCPs [47].

Overall, the findings of this study offer several actionable recommendations for healthcare organizations to address WPV. ED HCPs strongly endorsed interventions such as increased security presence, wearable devices like personal panic buttons, and WPV prevention training (Table 4). There was also significant support for improving debriefing processes and incident reporting systems, including implementing a call center to assist in completing reports. Additionally, understaffing emerged as a critical concern. The evidence supported many of these initiatives; however, the literature consistently highlights that implementing a single initiative is unlikely to be effective in addressing WPV [1,20,48]. While many of these initiatives require substantial funding, which may be a barrier for some organizations, the potential long-term benefits are noteworthy. The existing literature highlights the detrimental impact of burnout, WPV, trauma, and inadequate rest on HCP performance, including increased susceptibility to errors and reduced quality of care [7]. Investments in these interventions could yield significant long-term benefits, such as reducing turnover rates, improving workplace safety, and extending the careers of HCPs [7,22].

### 4.2. Limitations

This study has several limitations. The use of convenience sampling, while beneficial in facilitating participation among busy ED staff, may impact the generalizability of the findings due to potential biases associated with participant availability. Furthermore, our sample consisted exclusively of ED staff, with interviews conducted at two public hospitals in Toronto, ON, Canada. As a result, the findings may have limited applicability to hospitals in non-urban settings or other geographic regions, as well as private hospitals. The timing of data collection, conducted during the COVID-19 pandemic, is another important factor to consider, as the pandemic likely influenced HCP perceptions and increased WPV incidents while straining healthcare resources, potentially intensifying the challenges reported by participants.

## 5. Conclusions

This qualitative study contributes to the understanding of WPV in healthcare by applying a bottom-up approach that focuses on the perspectives of frontline ED HCPs rather than relying on quantitative data. The thematic analysis highlighted key insights from ED HCPs, offering valuable information to inform QI strategies within healthcare institutions while strengthening the generalizability of the findings from similar studies and introducing new findings. Additionally, this study established a baseline for QI efforts within our organization. Future rounds of interviews, conducted after the implementation of interventions, will assess the effectiveness of these initiatives from the ED HCPs’ perspective, further contributing to the literature on WPV in healthcare and providing evidence-based strategies for other healthcare institutions.

## Figures and Tables

**Table 1 healthcare-13-00679-t001:** Roles of the interview participants (n = 52). * Six interviews were subsequently lost due to technical errors, resulting in a final analysis sample of n = 46.

Role	Number *
Nurse	29
Psychiatric Emergency Service Unit Nurse	5
Physician	4
Medical Radiation Technician	3
Ward Clerk	3
Housekeeper	2
Security Guard	2
Social Worker	2
Learner	2

**Table 2 healthcare-13-00679-t002:** Summary of themes identified through thematic analysis, supported by representative quotes and the frequency of the themes.

Theme	Subtheme	Evidence
**1. Violence is Part of the Job (n = 43)**	1.1 Desensitization to WPV in the ED (n = 37)	“*I think it’s an interesting thing where we’ve had to just adapt to the reality that violence is like a normal part of your day. I’ve never been physically assaulted in the ED which I’m grateful for, but the risk is there and you hear of lots of situations so, you just kind of accept that potentially violent patients are here and we have to care for them.*” (Interview #17)
	1.2 A Prevailing Sense of Hopelessness (n = 32)	“*I think it just becomes normal in emerge now when a person is violent. That’s just normal. Not that I don’t think they’re [management] taking it seriously, but I just think it’s so busy now. No one has the time to do any debriefing, or counseling or following up on these incidents. So I think people just get left behind and their voice isn’t really heard.*” (Interview #28)
	1.3 Identity-Targeted in WPV (n = 6)	*“The psychological trauma of having patients berate you, call you names, use racial slurs, comments about people’s physical attributes, et cetera, like those things happen every single day and can be truly, truly traumatic.”* (Interview #53)
	1.4 Moral Injury and Moral Distress (n = 35)	“*We’ve had quite a few situations over the past few years that have caused people to not even want to work in this environment. It leads to burnout and then it spirals into this process where people end up quitting because of the repeated workplace violence. And that’s confounded by having to provide care for people who are sick. So when you’re trying to deal with providing that care in addition to somebody who’s aggressive and somebody who’s violent, it kind of compounds all together.*” (Interview #23)
**2. Leadership Dynamics regarding WPV (n = 32)**	2.1 Support from ED Management (n = 17)	“*I think they [immediate management] provide us with the support. Our leadership team has been pretty great and our security team has been really great in providing that support. I don’t think the training that we receive before we start is sufficient, especially with the psych patients. Not everyone has a psych background and not everyone knows how to interact and what might trigger a psych patient.”* (Interview #43)
	2.2 Perceived lack of Organizational Initiative with regards to WPV (n = 31)	“*You do an incident report. But what’s the point? Management is supposed to get back to you… But nobody ever gets back to you. There’s nothing, there are no solutions. And then we ask how can we make it better? Again, there’s nothing. And then it happens again. So then what’s the point in completing one?*” (Interview #41)
**3. Perceived Disconnect Among ED Staff During WPV Response (n = 41)**	3.1 Interactions between roles in the ED (n = 24)	“*It just felt like there’s way too many people around. When someone was escalating and then a lot of people showed up and like I said, I’m not really sure what they’re supposed to be doing. There’s one person that engages with the patient from what I had seen, so I’m not sure if that is the most beneficial in emerge when we already have a relationship with the patient.”* (Interview #38) “*I think it’s [the higher prevalence of workplace violence against nurses] because the nurses spend more time with patients in general. A doctor sees you for five minutes, but then we’re the ones that have to assess you, do the orders, keep talking to you and convince you to take the medication or whatever may be. So we have to interact with them more, I think it’s just the nature of our job.”* (Interview #14)
	3.2 Role Confusion during WPV Incidents (n = 35)	“*Security guards and nurses should work together during a code situation. However, sometimes we find that some security guards may take charge of situations and want to manage the Code White without really listening to the advice of nurses. It’s good for nurses to give their medical advice as well and take charge so that we can de-escalate and provide other interventions. If the guards are aware of that, then that’s better.*” (Interview #37) “*The other thing we do in a Code Blue that I have never seen done in a Code White situation is somebody identifying themselves as a team leader. In a Code Blue, when I walk into the room, I say my name is so-and-so and I’m one of the staff emergency physicians. I’m running this code so that people understand my role. But you never see a team lead identifying themselves in a Code White situation. You never see anybody identifying themselves in a Code White situation.*” (Interview #52)
**4. Enhancing Systems and Culture for Effective WPV Management (n = 46)**	4.1 Current Measures and Feelings of Safety (n = 40)	“*Overall, I think the guards are helpful. Our current one is obviously great, but two is getting towards being great just in terms of feeling safe and having someone around that can be like rougher with a patient, whereas we can’t.*” (Interview #2)
	4.2 Areas for Improvement and Prospective Measures (n = 46)	“*I think that [body-worn cameras] would be useful. It could help deescalate situations quicker if the patients see that they’re being recorded. Or staff think they’re being recorded. I think it works for the police well.*” (Interview #28) *“I feel like that’s [personal alarm systems] been helpful when I’ve worked on departments. I’ve worked on a locked psychiatric unit and everybody had them, and I feel like it was helpful to them. There are logistical things that go along with that, like false alarms and stuff. But I think for the number of times that it would be needed, it would be used and be functional.*” (Interview #42)
	4.3 Need for Education (n = 42)	“*Definitely NVCI (non-violent crisis intervention) and CPI (crisis prevention intervention) training would work. I’ve seen it work multiple times. I can even say I’ve seen it work on patients here and patients that are worse than what you might see as from a Code White here. So I’m a very strong believer in CPI and NVCI trainings if those courses are being done annually and are provided by UHN.*” (Interview #49)
	4.4 Streamlined Code White Process (n = 26)	“*Sometimes when security gets there [to a Code White], they usually arrive like one, then two and three, then four, then five and I don’t know if they have a system in place, but sometimes some of them are just standing there like a deer in headlights without kind of knowing what they do. But they do look to us for guidance a lot and sometimes it’s like, well, they’re throwing chairs screaming in a room threatening to punch me. What else can I tell you? I think having those people [Behavioural Emergency Response Team] for like the code white, like having a code white be treated just like a resuscitation would be good. Like someone is in charge of deescalating the situation, someone maybe in charge of trying to like physically calm the person. I think that having rules would be good.*” (Interview #14)
	4.5 Environmental Concerns (n = 30)	“*I noticed when a patient escalates unexpectedly, there are no security buttons in a couple of locations where I would like them to be. So, I think a re-evaluation of where the security buttons are, in particular, subacute one and two, don’t have them…If a patient’s blocking your entry or exit from the room, there’s no button inside the room or even just outside. I think that’s something that should be re-evaluated too.*” (Interview #40) “*When you enter to see the patients, the way that the beds are put, sometimes you find yourself on the opposite side of where the entrance is. So, if you were in that sort of a situation, there is no easy way for you to escape…If a nurse goes in and she’s for example, putting up leads to hook the patient up onto a cardiac monitor and if she was to be attacked by the patient, I can think of many of the areas and rooms around where she would not have easy access to be able to escape. Because she’d be locked or he’d be locked. You know what I mean? So those things have got to be thought out.*” (Interview #18) “*I think stupid little mistakes that could be serious ones. I found a scalpel next to the printer and when I brought it to the charge nurse, it was just kind of like, oh, it’s just a scalpel. And I’m thinking but if it didn’t get used here, a patient who is totally messed up, they could have taken it outside the hospital. Stupid little mistakes, I think could cause something like so serious like leaving a needle, laying around. Like, I certainly don’t want to be stabbed with that needle.*” (Interview #5)

**Table 3 healthcare-13-00679-t003:** ED staff perceptions of security guards.

ED Staff Perceptions of Hospital Security	Yes	No	Unsure	N/A
Security is part of the Medical Team	31	3	6	6
Security increases staff sense of safety	37	1	1	7

**Table 4 healthcare-13-00679-t004:** ED staff responses to proposed WPV prevention interventions.

Suggested Intervention	Yes	No	Unsure	N/A
Environmental Indicators for harm reduction	27	17	2	0
Personal Panic Alarms for ED Staff	43	2	1	0
Additional Security Guard in the ED	29	15	1	1
Wearable Devices for Security Guards	19	12	12	3
Flagging Patients with Behavioural Issues	42	1	0	3
De-Escalation Training	38	4	2	2
Code White Simulations	41	3	0	2
Code White Governance Committee	38	2	4	2
Updating Incident Reporting	37	2	4	3
Routine Check-in with ED Staff	40	2	0	4

## Data Availability

The thematic analysis and associated quantitative data supporting the findings of this study are included within the manuscript. Due to the sensitive nature of the qualitative data and to protect participant confidentiality, interview transcripts and coding files are not publicly available but can be accessed upon reasonable request to the corresponding author, subject to institutional review board approval and appropriate data-sharing agreements.

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
