# Peer review of "Exploring the Impact of Workplace Violence in Urban Emergency Departments: A Qualitative Study†"

_healthcare, 2025, doi:10.3390/healthcare13060679_

Round 1
Reviewer 1 Report
Comments and Suggestions for Authors
General comments:
· Thank you for the opportunity to review this important piece of scholarship on workplace violence in emergency departments.
· There are times throughout the manuscript where you swap between past and present tense. Please keep your tense consistent throughout.
Introduction:
· The introduction and problem formulation are very brief. I would like to see more detailed information here. For example, can you discuss prior research in more detail (method, sample)?
· You have justified your qualitative approach by critiquing quantitative research. It would be good to see you also discuss prior qualitative research on WPV in ED/ healthcare (findings, strengths, limitations) to better justify your study.
· Could you include a research question in the ‘purpose’ section?
Method:
· It’s great to see a section on researcher characteristics and reflexivity as this often isn’t done! Could you please expand this a bit more to include info about all authors involved in the analysis and to highlight how your prior experiences might have impacted the analysis?
· You have stated that you used a phenomenological approach so a better justification for your choice of analysis would be good. E.g., why not interpretive phenomenological analysis?
· To aid flow and reduce repetition, some subsections could be combined. E.g., context, sampling strategy, units of study; data processing and data analysis
· Braun and Clarke (2021) have differentiated between different types of thematic analysis: coding reliability approaches, reflexive approaches, and codebook approaches. Could you please specify which approach you have used and justify why it was chosen over others?
· The analysis can be described in more detail – what steps did you actually take to analyse the data? Did you do inductive or deductive coding? Latent or semantic coding?
Results:
· You have not provided any supporting data (illustrative interview quotes) in the results. This makes it hard for the reader to evaluate your analysis. It currently reads as a summary with no evidence to back up what you are saying. EDIT: I can see this at the end of the results in Table 4. It would flow better if quotes were integrated in-text or Table 4 was presented much earlier in the results.
· Tables 2 and 3 (or 3 and 4?) reads as though you also asked some quantitative yes/ no questions. If this is the case, this should be specified in the method section.
· The quantitative section needs to be detailed in the method. E.g., how did you come up with the list of interventions to ask about?
· Your interview questions (provided in the supplementary material) line up quite closely with your themes suggesting an under-developed analysis.
Discussion:
· The discussion (particularly the first half) reads similarly to the results section and more research could be integrated and discussed in more detail.
· “Additionally, ED HCPs with diverse intersectional identities reported experiencing verbal WPV directed at aspects of their identity” – this was not reported on in the results so why is it described in the discussion?
· Line 481 -if your study did not explore the effectiveness of peer support workers then I’m unsure why it is being discussed in such detail in the discussion.
Braun, V., & Clarke, V. (2021). Can I use TA? Should I use TA? Should I not use TA? Comparing reflexive thematic analysis and other pattern‐based qualitative analytic approaches. Counselling and psychotherapy research, 21(1), 37-47. https://doi.org/10.1002/capr.12360
Author Response
We thank the reviewer for their constructive feedback, please see the attached document with point by point feedback.

Reviewer 2 Report
Comments and Suggestions for Authors
This study aims to explore the impact of workplace violence on Healthcare Professionals working in Emergency Departments.
The study is one of the 12 subprojects as part of a larger study. This study interviewed 52 participants most of whom are nurses using an interdisciplinary team to perform the analysis.
Results is discussed under four main themes and subthemes under each of them. In these sections the authors have explained in text the information gleaned from the interviews but have not provided any actual numbers of responses to key questions. This should be explained in tables because without this it is difficult to evaluate the validity of the conclusions.
Table 2 provides the different roles that the participants played in the ED. This totals to 52 however in the tables 2 and 3 the total number of participants appear to be only 46 is there an explanation for this?
The main issue with this manuscript is that the results are not given in enough detail to warrant the discussion and conclusion. Here are examples of this so that the authors can revise the manuscript and provide more in-depth results section:
· “we will define WPV as any act of physical, verbal, psychological, and/or sexual violence committed by patients or visitors against HCPs”
It is important for the authors to lay the foundation by also providing information on the number of such incidents their severity, say over the past one year. Did participants each experience only one incident or several. Any data of severity leading to injuries did these have any bearing on the responses by the participants. This may help in triaging those who need quicker more intense intervention
Is there any difference among the incidents experienced by the nurses, physicians, social worker etc
· Do the responses differ among the various roles played. Is there a difference in response for those who experienced fewer incidents and those who experience more?
· The results of the responses need to be correlated with the type and severity of the incident and the number of incident each member of the staff may have faced, this could affect the answers that they provide to the interview. Since the majority of participants are nurses it is also important see if there's any difference between the responses by the nurse participants as opposed to the others.
· Under Leadership Dynamics Regarding WPV, manuscript does not give the data on how many said what?
· Vignettes from Code White: Sounding the Alarm on Violence Against Healthcare Workers were incorporated to prompt realistic discussions, with all receiving prior approval. This is not clear does it mean that workers affected in the code white were specifically interviewed after the incident. If so how soon after?
· 3.2 Violence Is Part of the Job:
Authors write: “This normalization is reinforced by the regularity of WPV incidents, including Code White events that often go unreported or are not formally called…” For this sentence to be acceptable it is important to know what is regularity of incidents and is there any classification of these incidents.
· What is the meaning of interdisciplinary role? Does this mean that a nurse may take another role. This should be explained better and numbers of how many participants were holding such roles need to be provided. How many had intersectional identities. If the actual response numbers are not given it is difficult to understand f the conclusion is warranted.
· “ED HCPs felt that the organization did not prioritize addressing WPV in the ED and that little initiative was being taken at the organizational level. Despite this, ED HCPs often viewed ED leadership as supportive, primarily due to observed actions taken within the ED and their physical presence and more visible ‘acts of solidarity’” . Where is the supporting data for this how many said what? Results with actual distribution of responses should be included in 3.3. Leadership Dynamics Regarding WPV maybe in table form
· “normalization of WPV within the ED” how many felt this, was it among all types of staff, those who had suffered more sever abuses or repeated abuses?
Finally, there are several papers published on workplace violence in Hospital settings, what is this manuscript adding to the existing knowledge? Discussion should emphasize this. Especially since some possible interventions were included in the interview questions, more detailed analysis of these responses can provide valuable concepts for other institutions
Author Response

(The authors gave the same response as above.)

Reviewer 3 Report
Comments and Suggestions for Authors
INTRO: Good description of workplace violence (WPV), health care providers (HCPs) and reasons a qualitative study was performed. The purpose of the study clearly explained - to explore ED HCPs’ lived experiences with WPV, their perceptions of support, safety, security and necessary changes needed to guide development of actionable interventions
MATERIALS + METHODS: Sampling strategy, ethical issues, data collection methods, data collection instruments and technology were well described. Supplements of the informed consent forms were attached at the end of the submission. Although not a large sample number, a broad range of professional providers in the ED were interviewed. Two researchers independently coded the data and discussed discrepancies until a consensus was reached. Several independent ED experts were consulted to ensure that emerging themes were consistent and valid with lived experiences.
RESULTS AND CONCLUSIONS: Four main themes were identified:(1) Violence is Part of the Job, (2) Perceived Leadership Dynamics regarding WPV, (3) Disconnect Among ED Staff During WPV Response, and (4) Enhancing Systems and Culture for Effective WPV Management.
Frequency of WPV and lack of follow-up lead to desensitization and normalization of WPV, skepticism regarding effectiveness of WPV preventive interventions, prioritizing patient care over one’s own safety, moral injury and moral distress leading to job burnout, acceptance culture discourages discussion and dealing with WPV as an organizational problem.
Perceived support from ED management was high, but there was a perceived lack of institutional organization support – a lack of communication, intervention, training and resources. HCPs reported strong teamwork. More MPV against nurses was felt to be related to increased time spent with patients and being female, more than about ED hierarchy. HCPs recognized importance and effectiveness of security personnel; however, security personnel did not feel included as part of the team or appreciated. There was miscommunication as to roles and responsibilities between HCP and security during an incident and there was a lack of a coordinated response to WPV events.
A stronger organizational culture and improved management support were seen as critical for enhancing staff safety and confidence in handling WPV; several specific points were noted in the results. Additionally, staff reported being severely understaffed, which hampered their ability to ensure both patient care and personal safety and additional security guards to support the ED team were needed.
Tables citing quotes from the interviews support the results noted above.
DISCUSSION: Good discussion integrating HCP perceptions of WPV and resulting mental health and wellbeing issues in HCPs along with findings in the literature. Clarification of perceptions and potential interventions (understaffing, education and training, additional security measures and personnel, environmental layout of ED) were discussed.
Although not addressed in this study, the use of peer support workers to improve communication between HCPs and marginalized patients, de-escalate potential WPV incidents and improved health care was discussed. Appropriate limitations of the study were noted.
Addressing just one of these themes would not be effective. All need to be addressed - even if this is costly to the institution. The improvement in worker retention, health and well-being of HCP is worth the cost.
I think this is a very valuable addition to the literature of health and safety of HCPs. The reader is not overwhelmed by statistics, rather there is a comprehensive discussion of the lived perceptions of HCPs, real problems in the ED and potential interventions. Of major importance, the discussion of peer support workers is very valuable. I was not aware of this and I suspect many of us in Health and Safety of healthworkers and hospital administration are not aware of what appears to be an effective and relatively affordable intervention to decrease WPV.
Author Response

(The authors gave the same response as above.)

Round 2
Reviewer 1 Report
Comments and Suggestions for Authors
Thank you for your thoughtful revisions and the opportunity to review a revised version of this manuscript
Introduction:
- Please provide a reference for your definition of WPV
- Please expand on your definition of HCP e.g., can you provide some examples of HCPs. What kind of health care are they providing, to whom, and in what context?
- “WPV in healthcare is a complex and multifaceted issue influenced by numerous risk factors, which can be categorized as clinical, environmental, organizational, societal, and economic” – can you expand on this. Perhaps provide some examples of the clinical, environmental and organizational risk factors. Could you also bring in a systems thinking theory here? This could add to your analysis, especially as you talk again about systemic factors in the discussion.
- “Prior qualitative studies on WPV in healthcare and ED settings advanced our understanding of WPV from HCPs’ perspectives, identifying contributing factors and unmet needs of HCPs [12–14].” – can you please expand here? What are the perspectives of HCPs on WPV? What contributing factors have been identified?
- You have said that a strength of your study is that it is longitudinal. Were any of the prior qualitative studies you mentioned [12-14] longitudinal? Perhaps this is something you can incorporate into your critical analysis.
- “Nevertheless, recurring themes across studies strengthened the broader understanding of WPV, while introducing new insights that may be institutionally unique contributes to furthering our understanding of WPV.” – this is quite vague and needs to be more specific.
Method:
- Thank you for the describing researcher characteristics. However, I do not feel that this information is suitable for a table. A few sentences would suffice. Could you expand to state how your collective experiences (e.g., exposure to WPV, clinical or non-clinical experience) may have impacted your approach to data analysis?
- You have stated that you have used reflexive thematic analysis however your described analysis suggests otherwise. For example, you have not referenced Braun and Clarke who developed reflexive thematic analysis, and you refer to saturation and inter-rater reliability which are concepts Braun and Clarke have renounced. It sounds as though you are using a different type of thematic analysis (e.g., template analysis). Please see the reading I suggested previously, and state which type of thematic analysis you have used (coding reliability approach or codebook approach). Similarly, Braun and Clarke are very clear that themes are generated through active engagement with the data not ‘identified’ or ‘found’.
- In the results, you state “HCPs who identified as men” but have not referred to demographic data like gender or age in the methods. Please provide this data if you have it.
Results:
- Please provide an in-text overview of the quantitative findings. For example, you could state which intervention had the most support and which one had the least etc.
Discussion:
- Again, the first 5 paragraphs of the discussion largely mirror the results and include very limited discussion of other research. Please revise further.
Author Response

(The authors gave the same response as above.)

Reviewer 2 Report
Comments and Suggestions for Authors
This paper has been improved by expanding on the Introduction sections especially lines 97-113.
Lines 231 -239 good so the source is explained as Code White calls
Addition of table 1 is not particularly helpful and does not add much to the paper and results.
250 -252 the table 2 should be modified to only show 46 interviews and not cause confusion with the total sample size.
One of the comments provided by the reviewer was that there is not enough details regarding the prevalence of responses in the four themes and subthemes, this is not sufficiently addressed. While table 5 gives quantitative information about what the intervention preference is for the participants, this is not available for the other themes.
Table 3 is an improvement and moving it before the textual explanations is a good way to present the results.
Each of the four main themes should have quantitative information of the responses. The semi structured interviews will be able to provide some information on the prevalence of responses. Without knowing how many participants gave what response how can the discussion be corroborated?
Example: table 3: “I think it's an interesting thing where we've had to just adapt to the
reality that violence is like a normal part of your day. I've never been
physically assaulted in the ED which I'm grateful for, but the risk is
there and you hear of lots of situations so, you just kind of accept that
potentially violent patients are here and we have to care for them.”
(Interview #17)”
How can the reader be sure that the author is not choosing a quote that appeals to them. As a reader we need to know what the prevalence was, how many people said something that follows this type of sentiment, and then Verbatim courts can be used to emphasize that quantitative information
Good examples of such quantitative information includes: Perceived Disconnect Among ED Staff During WPV Response, has one table 4 to explain about the security guards. Authors can also consider using charts like bar or pie charts to visualize the overall results and then delve into some notable verbatim responses.
Discussion line 583: A prevalent theme among participants was the normalization of WPV in the ED, paired with a perceived lack of organizational response. Which part of the results indicate this the prevalence?
The authors mentioned that this study is to understand the situation so that good interventions can be provided and good interventions are provided in Table 5 based on the quantitative analysis basis of their responses. This type of quantitative analysis must be provided for all the different sub themes and things
My suggestion is as follows for this to be published and good readable material authors have identified 4 main themes and sub-themes. The quantitative information about who answered what needs to be provided at a glance so that readers can see what the prevalence is for each of these teams and once that is done then they should give the quotable quotes to emphasize what they have understood from the quantitative numbers.
Author Response

(The authors gave the same response as above.)
